# Regional Drought Conditions Control *Quercus brantii* Lindl. Growth within Contrasting Forest Stands in the Central Zagros Mountains, Iran

Ehsan Safari [1], Hossein Moradi [1], Andrea Seim [2,3], Rasoul Yousefpour [4,5], Mahsa Mirzakhani [1], Willy Tegel [2], Javad Soosani [6] and Hans-Peter Kahle [2,*]

1  Department of Natural Resources, Isfahan University of Technology, Isfahan 84156-83111, Iran; safari.ehsan71@gmail.com (E.S.); hossein.moradi@iut.ac.ir (H.M.); mahsamirzakhani72@gmail.com (M.M.)
2  Chair of Forest Growth and Dendroecology, Institute of Forest Sciences, Albert-Ludwigs-University Freiburg, 79106 Freiburg, Germany; andrea.seim@iww.uni-freiburg.de (A.S.); tegel@dendro.de (W.T.)
3  Department of Botany, University of Innsbruck, 6020 Innsbruck, Austria
4  Institute of Forestry and Conservation, John Daniels Faculty of Architecture, Landscape and Design, University of Toronto, 22 Ursula Franklin Str., Toronto, ON M5S 3H4, Canada; rasoul.yousefpour@ife.uni-freiburg.de
5  Chair of Forestry Economics and Forest Planning, Albert-Ludwigs-University Freiburg, 79106 Freiburg, Germany
6  Faculty of Agriculture and Natural Resources, Lorestan University, Khorramabad 68151-44316, Iran; soosani.j@lu.ac.ir
*  Correspondence: hans-peter.kahle@iww.uni-freiburg.de

**Abstract:** The magnitude and duration of ongoing global warming affects tree growth, especially in semi-arid forest landscapes, which are typically dominated by a few adapted tree species. We investigated the effect of climatic control on the tree growth of Persian oak (*Quercus brantii* Lindl.), which is a dominant species in the Central Zagros Mountains of western Iran. A total of 48 stem discs was analyzed from trees at three sites, differing in local site and stand conditions (1326 to 1704 m a.s.l.), as well as the level and type of human impact (high human intervention for the silvopastoral site, moderate for the agroforestry site, and low for the forest site). We used principal component analysis (PCA) to investigate the common climatic signals of precipitation, air temperature, and drought (represented by SPEI 1 to 48 months) across the site chronologies. PC1 explains 83% of the total variance, indicating a dominant common growth response to regional climatic conditions that is independent of the local environmental conditions (i.e., forest stand density and land-use type). Growth–climate response analyses revealed that the radial growth of *Q. brantii* is positively affected by water availability during the growing season (r = 0.39, p < 0.01). Precipitation during April and May has played an ever-important role in oak growth in recent decades. Our study provides evidence that hydroclimatic conditions control tree-ring formation in this region, dominating the effects of topography and human impact. This finding highlights the great potential for combining historical oak samples and living trees from different forest stands in order to generate multi-centennial tree-ring-based hydroclimate reconstructions.

**Keywords:** tree-rings; dendroecology; principal component analysis (PCA); agroforestry; *Quercus brantii*; Iran

## 1. Introduction

Climate change is considered a crucial challenge in the biosphere, which can both positively and negatively impact forest ecosystems [1,2]. However, such projections are accompanied by uncertainties regarding the future magnitude and possible consequences of rapid environmental changes in dynamic ecosystems [3,4]. Having profound knowledge of forest growth is crucial to understand tree–environment interactions [5]. Evaluating the effects of climate change on forests is indispensable for semi-arid areas [6], such as the

Central Zagros Mountains situated in the Irano-Anatolian global biodiversity hotspot [7]. Species-specific growth responses must be considered in the assessment of climate change impacts on forest ecosystems [8]. With an average length of 1300 km and width of 200 km, the Zagros forests stretch from the northwest to the south of Iran [9]. The Zagros forests are a global biodiversity hotspot [7] and are characterized by a semi-arid Mediterranean climate with hot and dry summers and cool and moist winters. However, these forests are under severe pressure due to different disturbances, such as fire and drought, extreme overexploitation due to grazing and wood use for charcoal production, as well as due to clearing for mining and infrastructure construction [10].

The ring-porous tree species *Quercus brantii* Lindl. is the dominant species in the semi-arid oak forests of Lorestan in the Zagros Mountains in western Iran. This indigenous, deciduous, broad-leaved species covers an area of $2.2 \times 10^6$ hectares (ha) in the Zagros forests [11] and grows at altitudes between 450 m and up to 2700 m above sea level (a.s.l.) in Iran. Native to western Asia and with more than 50% coverage of the Zagros forests in Iran, it is one of the ecologically and economically most important tree species [12]. Semi-arid forests in the Zagros Mountains cover a vast area of six million ha and contain up to 44% of Iran's forests [13]. Several studies have analyzed the climate sensitivity of the tree-ring width of *Q. brantii* Lindl. In Iran [14–19]. As a result, precipitation reconstructions for the Zagros Mountains were developed by Azizi et al. [20], Jalilvand et al. [21], and Arsalani et al. [22]. Furthermore, wood anatomical characteristics of *Q. brantii* have been studied by [23,24], and the spatial distribution patterns of the species were studied by [9,11,25–27]. Despite the numerous dendrochronological studies on oaks in the Zagros Mountains [28–31], the effects of specific local site and stand conditions associated with different land-use types on the climate–growth relationships of *Quercus brantii* have not been investigated thus far. Moreover, different, or even contrasting, findings concerning the dominant climatic drivers of *Q. brantii* growth, i.e., responses to precipitation [14,17] and air temperature [16,20], were found in Central Zagros. Such inconsistencies may possibly be the result of differences in the growth response due to differences in the local site and stand conditions. As tree growth is affected by both the natural environmental conditions and human activities [32–34], human influences on forest ecosystems, and subsequently its impacts on climate–growth relations, ought to be studied in more detail [35]. Recent endeavors have attempted to attract more attention to the possible legacy effects of forest management systems on the climate–growth relationship. For instance, Mausolf et al. [36] suggested that former land-use partly affected the European beech trees' sensitivity to climate extremes in Northern Germany. Rodriguez-Vallejo and Navarro-Cerrillo [37] reported contrasting drought responses of *Pinus pinaster* in natural and planted forests in southern Spain. Another study in Germany proved the higher sensitivity of trees to drought in managed stands compared to unmanaged stands [38]. The legacy effects of forest management systems on forest dynamics in the Zagros Mountains have not been well studied yet. Alternatively, the few conducted studies came up with contrasting findings. For example, Valipour et al. [39] reported a decline in the tree regeneration rate due to the traditional silvopastoral management systems in Northern Zagros, while, according to Zabiholahii et al. [40], such management systems lead to an increased rate of tree regeneration in the referenced region.

Hence, knowledge of the effects of site factors (i.e., topography and elevation) and stand factors (i.e., forest stand density as the result of different forest usages) on the climatic response of *Q. brantii* is essential. In this research, we attempt to fill this gap for the Central Zagros Mountains and aim to answer the following questions: (1) how similar is the radial tree growth of *Q. brantii* between sample sites with differing site and stand factors, (2) what are the dominant climatic factors controlling *Q. brantii* growth, and (3) how stable is the climatic signal under changing climatic conditions?

## 2. Materials and Methods

### 2.1. Study Area and Sampling Sites

We selected three study sites (Shahanshah, Mele-Shabanan, and Dadabad) that share similar climatic conditions but differ notably in local site conditions as well as past and actual forest usages (i.e., silvopastoral, agroforestry, and forest) (Figure 1; Table 1). The study sites are located in the Khorramabad forests of Lorestan Province in Central Zagros (Figure 1). Silvopastoral and agroforestry are the major land-use management systems in the oak forests in the Zagros Mountains. At the silvopastoral site at Shahanshah, intensive grazing by livestock (sheep and goats) is the main management activity. The site is characterized by steep slopes with intensively eroded soils that show exposed rocks and tree roots. Compared to the other two sites, the density of the forest stand is the lowest. Rainfed agriculture is the main management system of the agroforestry site at Mele-Shabanan. The soil is less degraded than at the silvopastoral site. The forest site is located at Dadabad and is characterized by low management intervention (almost no wood extraction). This site has a higher forest stand density compared to the other two sites, and the soils show no signs of degradation. The study sites are between 20 to 60 km apart from each other and are administrated by the National Organization for Forests, Rangelands, and Watershed Management.

**Table 1.** Site characteristics including stand density, geographic coordinates, and elevation.

| Management Type | Site Name | Tree Density (n/ha) | Long. I | Lat. (N) | Elev. (m a.s.l) |
|---|---|---|---|---|---|
| Silvopastoral | Shahanshah | 40 | 48°09′02″ | 33°13′47″ | 1490 |
| Agroforestry | Mele-Shabanan | 77 | 48°09′16″ | 33°31′20″ | 1326 |
| Forest | Dadabad | 92 | 48°13′49″ | 33°19′07″ | 1704 |

### 2.2. Sampling and Chronology Development

A total of 48 stem discs (one from each tree) was sampled from *Q. brantii* coppice trees at the three sites (Table 2). The discs were collected from trees that were felled by Lorestan's Department of Natural Resources between 2011 and 2015 to prevent the spread of oak dieback (induced by insects and fungi) to unaffected areas and, thus, to protect healthy oak trees in those forests. The stem discs were taken at breast height using a chainsaw. In the laboratory, the samples were air-dried, and the cross-sectional area was sanded with successively finer sandpaper and, before the measurement, moistened with 90% ethanol to increase contrast for better visibility of the tree-ring boundaries [17].

**Table 2.** Chronology characteristics including number of sample trees and measurement radii, period length, mean segment length (MSL), average growth rate (AGR), minimum and maximum average growth rates, mean sensitivity (MS), and mean inter-series correlation coefficient (Rbar).

| Management Type | Number of Trees | Number of Radii | Chronology Time Span | Length of Chronology | MSL (Years) | AGR (mm/Year) | Min AGR | Max AGR | MS | Rbar |
|---|---|---|---|---|---|---|---|---|---|---|
| Silvopastoral | 22 | 64 | 1961–2011 | 51 | 36 | 3.05 | 1.49 | 6.68 | 0.342 | 0.595 |
| Agroforestry | 14 | 40 | 1959–2015 | 57 | 43 | 2.77 | 1.57 | 5.32 | 0.361 | 0.593 |
| Forest | 12 | 37 | 1948–2014 | 67 | 52 | 2.61 | 1.05 | 6.01 | 0.325 | 0.561 |

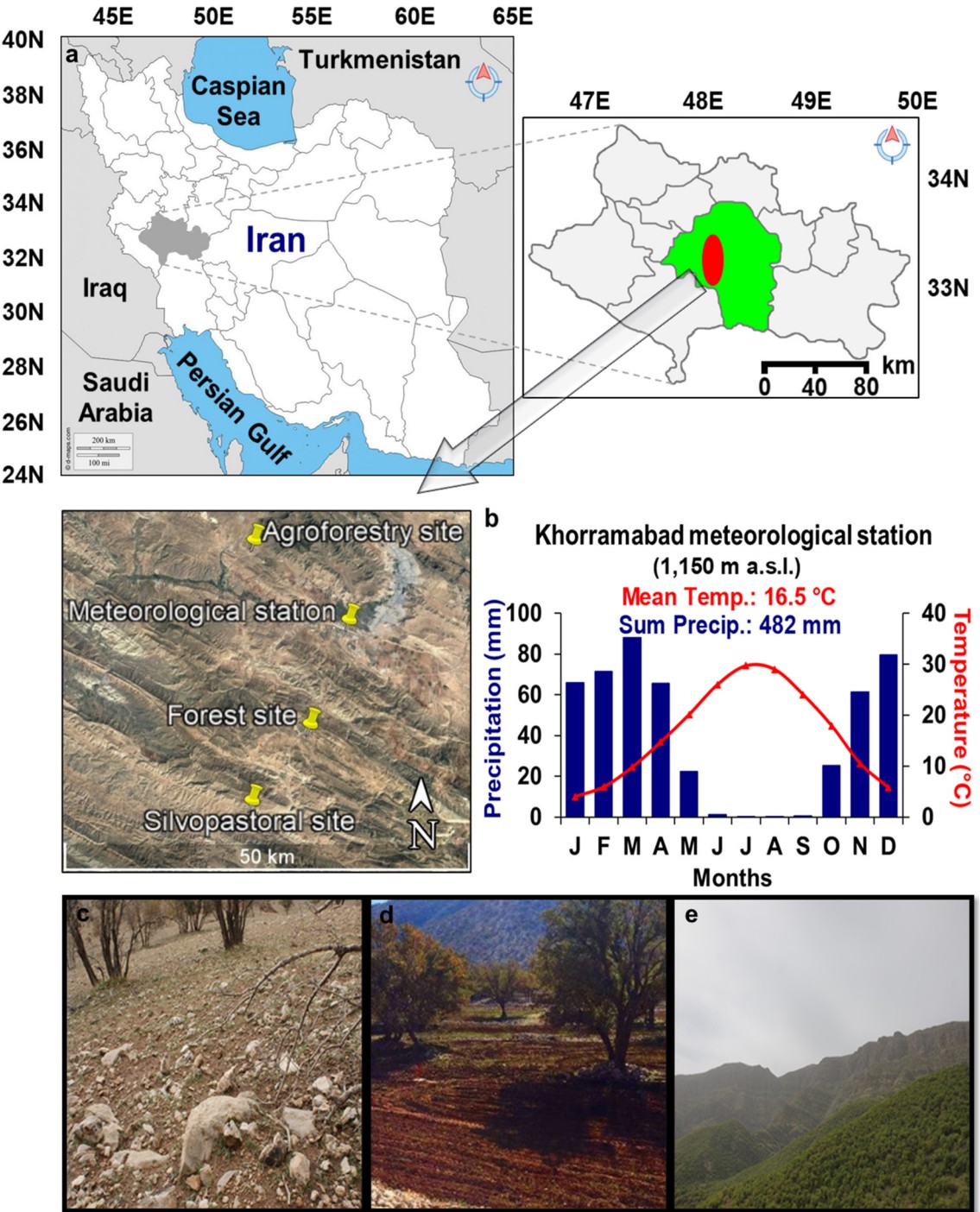

**Figure 1.** (**a**) Location of the three sampling sites and Khorramabad meteorological station within eht Central Zagros oak forests (red oval), Khorramabad (green area), Lorestan (grey area) in Iran; (**b**) climate diagram that shows average monthly precipitation sums (blue bars) and average monthly temperature means (red line) at Khorramabad meteorological station for the period 1982–2011; (**c**) Shahanshah site with silvopastoral system including intensive livestock grazing; (**d**) Mele-Shabanan site with agroforestry system of rainfed agriculturI (**e**) Dadabad site with forest system and low human intervention.

Each disc was scanned with high resolution, and the annual tree-ring width (TRW) was measured using the semi-automatic software system WinDENDRO (http://www.regentinstruments.com, accessed on 15 September 2019). Four equidistant radii per cross-section were measured and inter-radii angles were adjusted in the case of ingrown branches or wood decay. After automatic detection of tree-ring boundaries, the measurements of each radius were visually checked for missing or false rings. Individual measurements were imported into the PAST4 software (http://www.sciem.com, accessed on 15 September 2019) for synchronization and cross-dating of the TRW series. This process was done by considering the statistical parameters of mean synchronicity (sign test) and *t*-test after Baillie and Pilcher [41], along with on-screen visual TRW curve comparisons. This procedure was applied to the TRW series from all radii per disc and among discs.

The final verification of the cross-dating accuracy and assessment of possible measurement errors was done using the computer program COFECHA [42,43]. To remove age-/dimension-related growth trends, the raw TRW series for each site (i.e., the measurement series of each individual radius) was converted into dimensionless indices after applying a cubic smoothing spline with a 50% frequency cut-off of 30 years to emphasize high-frequency variability by employing the program ARSTAN [44]. The dendrochronological statistics of the mean inter-series correlation (Rbar), mean sensitivity (MS), and expressed population signal (EPS) [45] were calculated to characterize and evaluate the quality of the site-specific TRW chronologies. For the decomposition of orthogonal signals in the multivariate set of site-specific TRW chronologies, we applied a principal component analysis (PCA) [46] to the normalized series for the common period 1963–2011.

### 2.3. Climate Data

Monthly air temperature and precipitation data for the period 1963–2011 were obtained from the Khorramabad meteorological station (33°26′ N, 48°17′ E; 1150 m a.s.l.), which is located 27 to 50 km away from the sampling sites and has been in operation for almost 70 years (Figure 1a). These data are the most reliable climate data available for our study area, considering it is the nearest climate station to our sampling area. The baseline annual precipitation sums and temperature means in the study area are 482 mm and 16.5 °C, respectively. As semi-arid climate conditions in this region prevail, precipitation is low during summer (June to August) and highest during winter (previous year December to current year February), as well as during spring (March to May) (Figure 1b). Additionally, the standardized precipitation evapotranspiration index (SPEI) [47] was extracted from KNMI Climate Explorer (http://climexp.knmi.nl, accessed on 15 September 2019) as a spatial average for the region 47°8′–49°8′ E and 32°13′–34°13′ N and for different monthly means ranging from 1 to 48 months. The SPEI is a drought index that has been widely used in dendroecological studies [48–55] and varies from negative to positive values. The negative values represent dry climate conditions, i.e., water deficit, and vice versa. Trends in climate variables were assessed using the Mann–Kendall test [56] over the common 1963–2011 period. Annual variability and temporal trends of the mean air temperature and precipitation sum are shown in Figure 2. The mean annual temperature showed a decreasing trend over 1963–1983 and an increasing trend for 1984–2011. The Mann–Kendall test exhibited significant negative changes in the annual precipitation and mean annual temperature from 1963 to 2011 (Table A1). It also showed significant negative changes in the annual SPEI at 36- and 48-month scales. Significant negative monthly trends were detected for temperature (January, March, April, and from September to December), May precipitation, SPEI 24 (November and December), and from January to December for SPEI on a 36- and 48-month scale. For the climate–growth response analysis, in addition to the monthly values, the seasonal temperature and SPEI means and precipitation sums were calculated, including previous year December–February (DJF), March–May (MAM), June–August (JJA), September–November (SON), previous year December–May (D–M), April–September (A–S), and March–October (M–O).

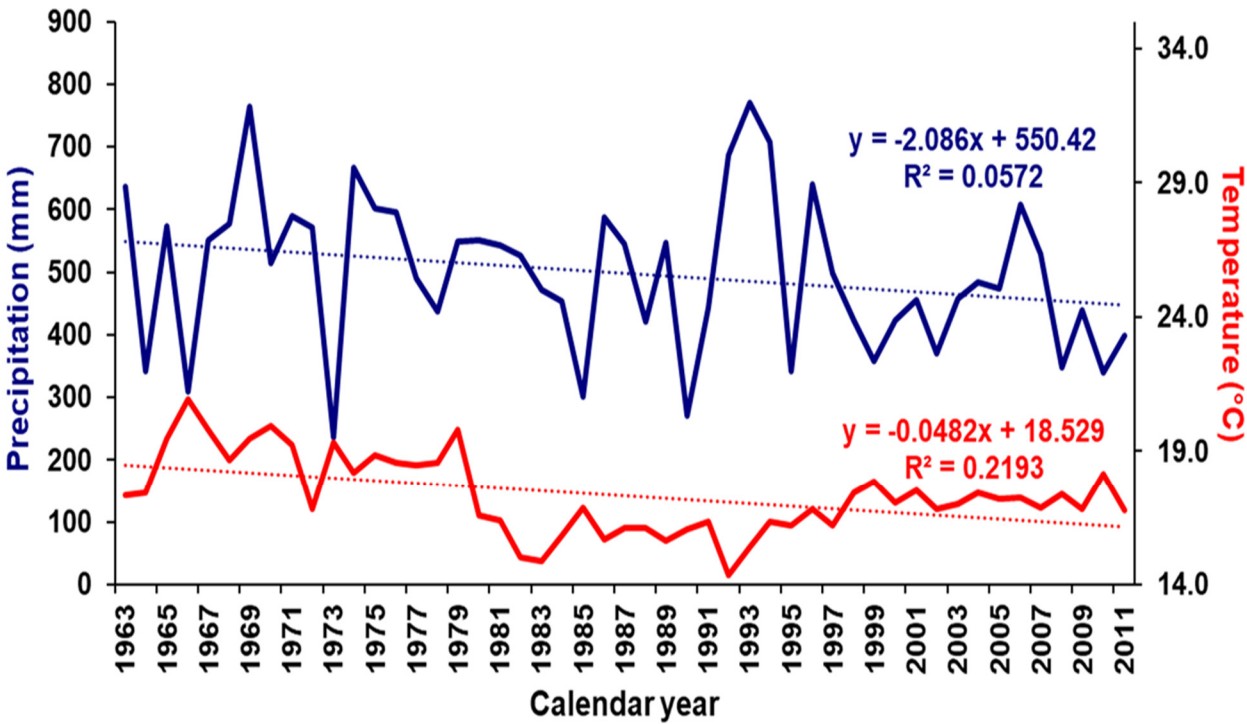

**Figure 2.** Total annual precipitation (blue line) and mean annual air temperature (red line) at Khorramabad meteorological station from 1963 to 2011. Dashed lines represent linear trends.

*2.4. Climate–Growth Response*

Pearson correlation coefficients were calculated between principal components (PCs, with focus on PC1) and the three climate parameters of air temperature, precipitation, and SPEI for the individual months (starting with previous year January to current year December) and seasons for the common 1963–2011 period. In addition, heat-map charts were generated to track the response to drought with an increasing time scale, i.e., from one month to two years, and also to assess the effects of precipitation and temperature on the radial tree growth. To summarize the hydroclimate signals in the three PCs (i.e., PC1, PC2, and PC3) over the growing season, we plotted radar charts. We tested the temporal stability and consistency of the climate–growth relationships (i.e., the correlations between total monthly precipitation, mean monthly temperature, and drought with PC1) by moving the correlation intervals of 20 years overlapping by one year using the "treeclim" R-package [57]. We chose January to December and the common 1963–2011 period. To perform the moving correlations between drought and TRW chronologies, the SPEI08 was selected as it revealed the highest correlation coefficients.

Spatial correlation maps were generated using the KNMI climate explorer (http://climexp.knmi.nl, accessed on 15 September 2019) and CSIC SPEI 08 and CRU TS4.04 temperature and precipitation grid data (0.5° × 0.5°) for the period 1963–2011.

### 3. Results

*3.1. Chronology Characteristics*

The mean segment length, equivalent to the average number of tree-rings at breast height, of the TRW records varies from 36 years at the silvopastoral site to 52 years at the forest site (Table 2). The highest average growth rate (AGR) was documented for the site with silvopastoral management system, while the lowest for the forest site. Likewise, the highest individual AGRs were also recorded for the silvopastoral site for segment lengths from 23 to 51 years, while the lowest individual AGRs for the forest site for segment lengths from 38 to 61 years (Figure 3).

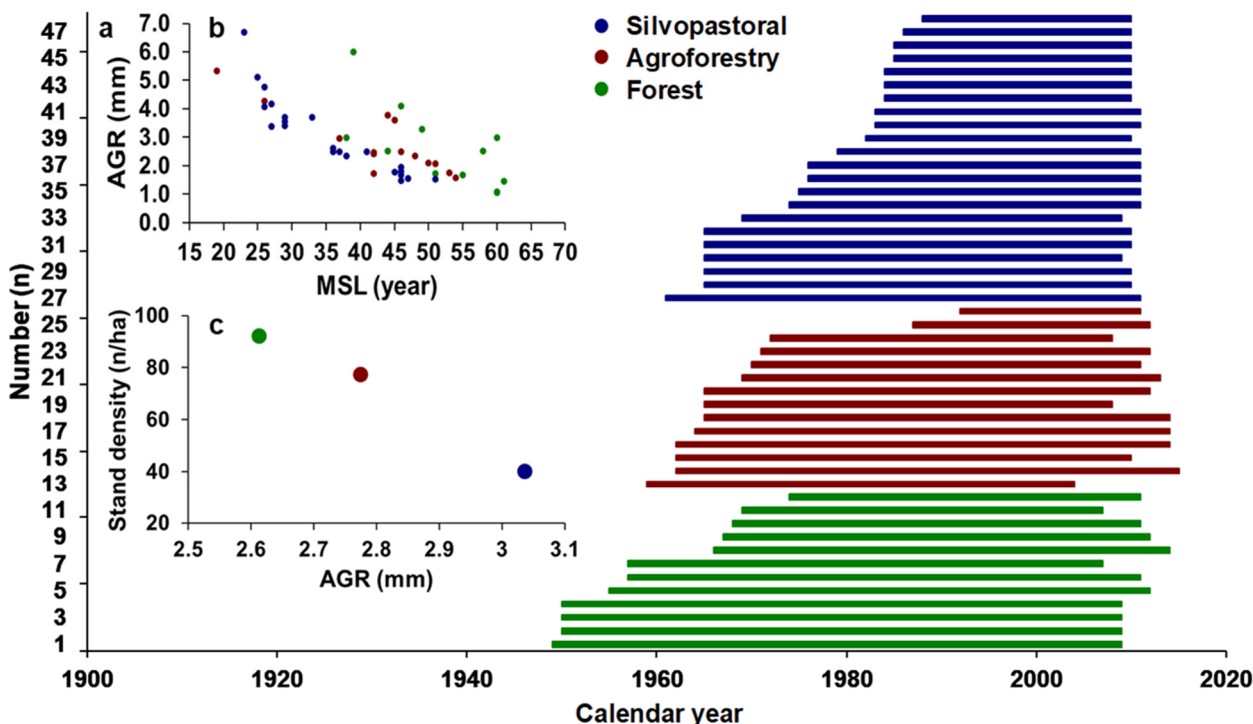

**Figure 3.** Temporal distribution of: (**a**) the 48 individual *Quercus brantii* Lindl. samples for the three sites; (**b**) relationship between average growth rate (AGR in mm/year) and mean segment length (MSL in years) of each sample; and (**c**) the relationship between stand density (n/ha) and average growths rate for each site.

For the forest TRW chronology, the EPS threshold of 0.85 was exceeded after 1960, as well as after 1964 and 1966 for agroforestry and silvopastoral, respectively (Figure 4b). The MS of the agroforestry chronology was higher compared to the other sites, which indicates a higher inter-annual variability of radial growth under this treatment type. Alternatively, the lowest MS was obtained for trees from the forest site. The Rbar varied from 0.561 (forest site) to 0.595 for the trees managed under the silvopastoral system. The three developed TRW chronologies showed very strong, significant positive correlations ($p < 0.01$) between one another for the common 1963–2011 period (Figure 4b, the years 1961 and 1962 were omitted because of large deviations, most probably due to juvenile growth in the agroforestry chronology).

Evaluating the age-DBH relationship, we found a significant negative relationship between tree age and stem diameter at breast height for the trees at the silvopastoral site (Figure 4a).

Results of the PCA showed that the first PC (PC1) explains 83% of the common variance, PC2 explains 10%, and PC3 explains 7% (Figure 5). All three chronologies have positive loadings on PC1, ranging from 6.19 for silvopastoral to 6.43 for the forest chronology. PC2 differentiates between negative loading for forest and agroforestry and positive loading for silvopastoral. PC3 differentiates between negative loading for agroforestry and silvopastoral and positive loading for forest.

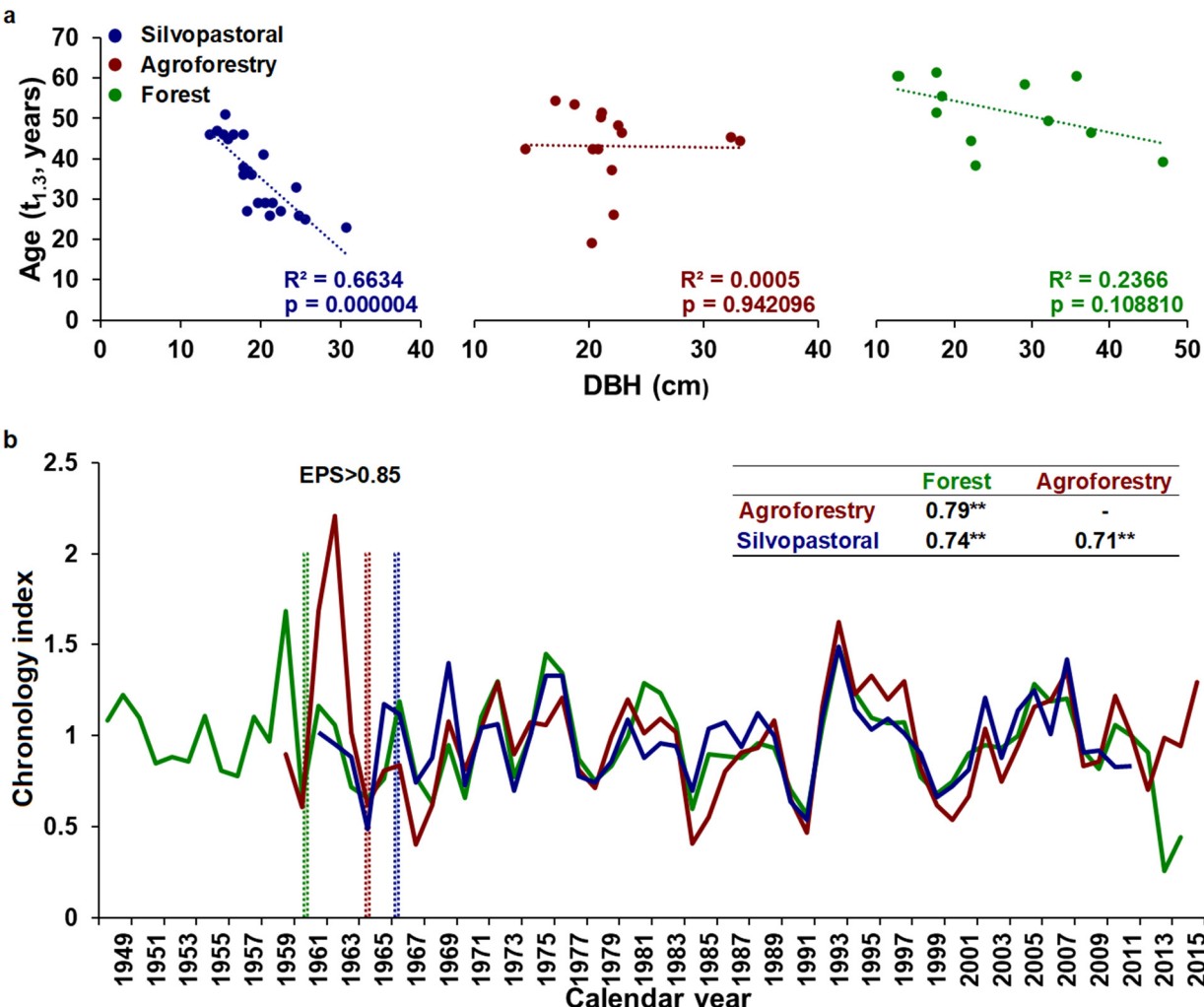

**Figure 4.** (**a**) Relationship between diameter at breast height (DBH, cm) and tree age (t$_{1.3}$, number of tree-rings at breast height) for all the three sites; (**b**) site-specific TRW chronologies for silvopastoral (Shahanshah, blue line), agroforestry (Mele-Shabanan, red line), and forest (Dadabad, green line). The vertical dashed lines indicate the first year with expressed population signal (EPS) > 0.85. The inset table shows the correlation coefficients between the three chronologies from years 1963–2011; the 99% significance level is denoted by two asterisks.

### 3.2. Growth–Climate Relationships

Figure 6 displays the climate–growth relationships between PC1 with temperature, precipitation, and drought index (SPEI 01 to SPEI 48) for the period 1963–2011. With precipitation, significant correlations (positive) were obtained for April (r = 0.49, *p* < 0.01) and July (r = 0.40, *p* < 0.01), for the winter (r = 0.50, *p* < 0.01) and spring (r = 0.50, *p* < 0.01) seasons, and short (r = 0.39, *p* < 0.01) and long (r = 0.50, *p* < 0.01) vegetation periods. No significant correlations were found for temperature. The results indicate that high temperatures, especially during April and spring, tended to negatively affect oak growth.

The highest significant positive correlation coefficients from SPEI 03 to SPEI 24 from January to October and most seasonal means were obtained. Furthermore, from SPEI 03 to SPEI 08, the correlation coefficients decreased towards the end of the year and towards longer SPEI means. Regarding seasonal means, the responses to hydroclimatic conditions during spring and for the growing season are strongest for all temporal scales of the SPEI.

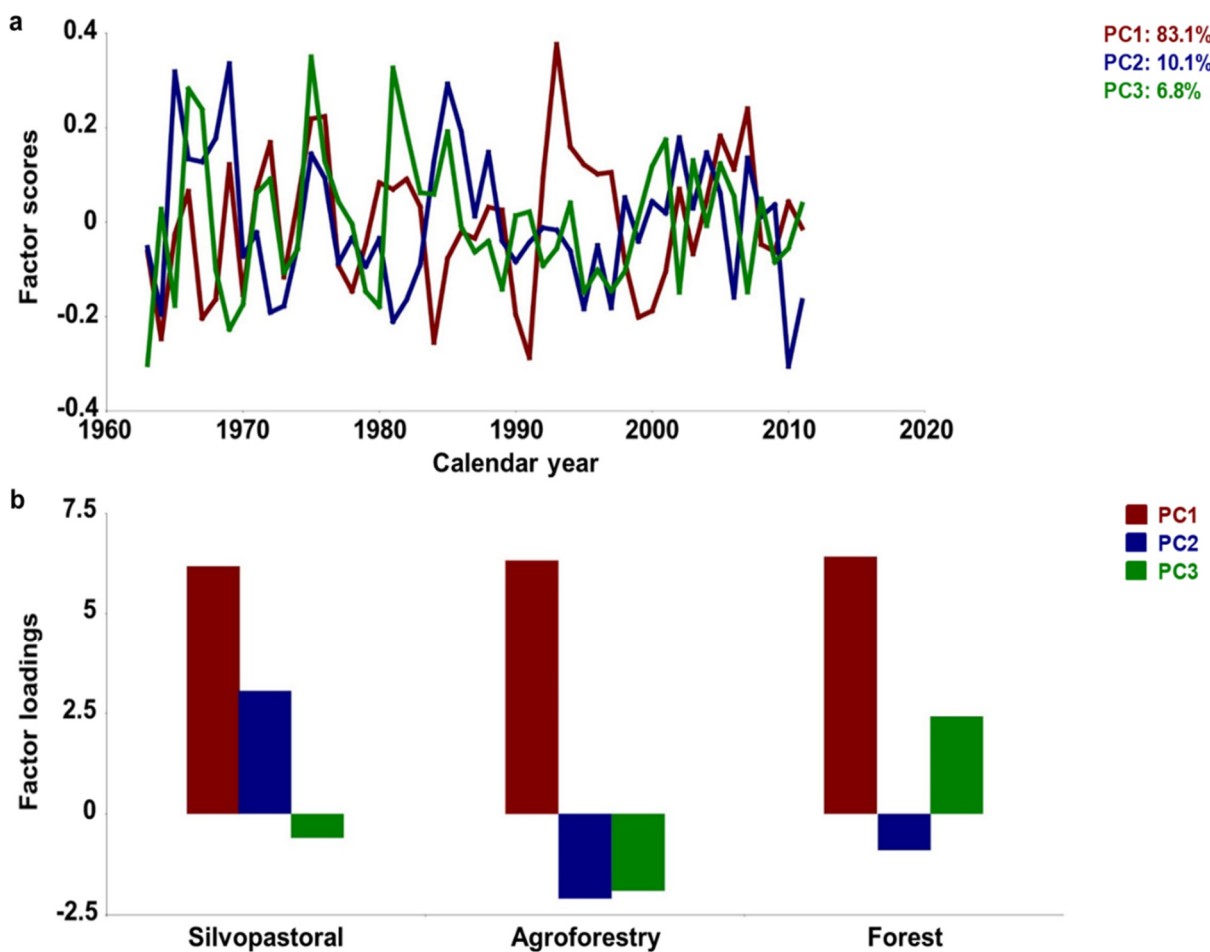

**Figure 5.** Results of the principal component analysis (PCA) for three TRW chronologies: (**a**) The line graph shows the time courses of the factor scores of PC1, PC2 and PC3; (**b**) the bar graph shows the factor loadings on PC1, PC2, and PC3 for the three chronologies.

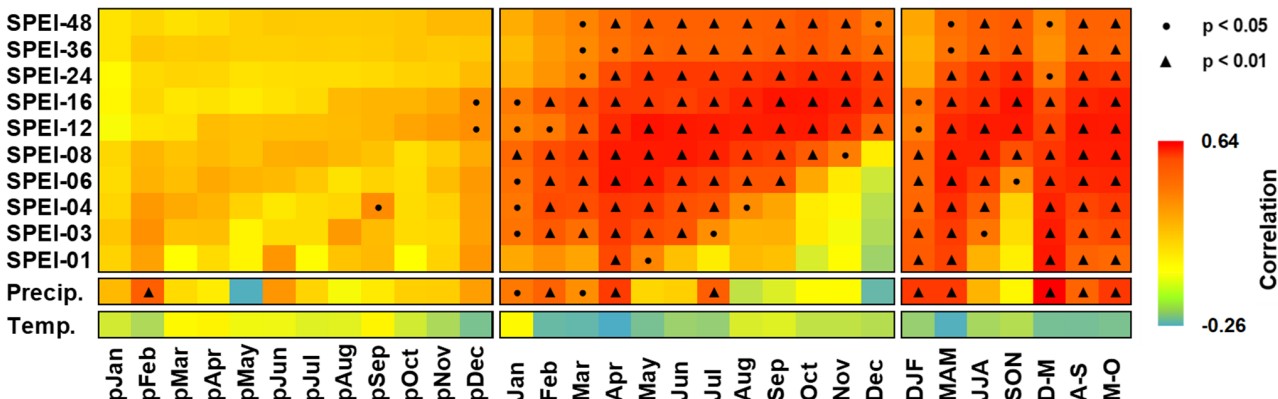

**Figure 6.** Climate–growth relationships (Pearson correlation coefficients) between the first principal component (i.e., PC1) and temperature, precipitation, and drought index (from SPEI 01 to SPEI 48) over the period 1963–2011. The correlation analysis applied for January of the previous year (pJan) to current year December and various seasons (DJF = previous year December–February (winter), MAM = spring, JJA = summer, SON = fall, D–M = previous year December–May, A–S = April–September, and M–O = March–October).

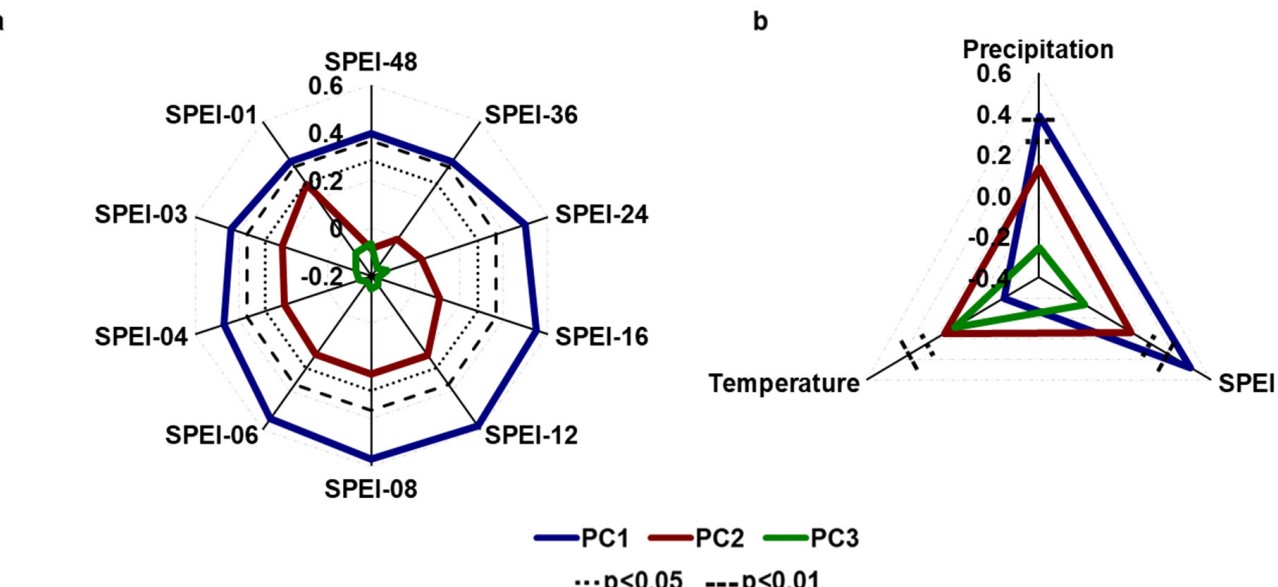

**Figure 7.** Comparison of relationships between the principal components (i.e., PC1, PC2, and PC3) and climatic variables: (**a**) mean April–September SPEI from 1- to 48-month time scales, and (**b**) April–September means of precipitation, SPEI (averaged of all April–September means), and temperature (see legend for colours).

For PC2 and PC3, no significant correlations are obtained. Hydroclimate conditions during April–September, expressed in the pronounced precipitation (r = 0.39, *p* < 0.01) and drought (SPEI; r = 0.48, *p* < 0.01)) signal (Figure 7b), over one to even two years strongly affected oak growth in the Zagros Mountains. In contrast, there is no noticeable relationship between temperature and growth over the mentioned timescale for all three PCs.

*3.3. Temporal Stability and Spatial Extent of the Hydroclimate Signal*

The temporal stability of the identified hydroclimatic signal was investigated through a moving correlation analysis which showed a significant and stable precipitation signal for April and May from the late-1960s. Additionally, significant positive correlations since the 1960s were found for January, February, and September, which, however, began to fade from the mid-1970s onwards. For the remaining months, no clear and stable relationship to precipitation was found (Figure 8). For SPEI 08, highly positive correlations were obtained from April to September indicating a persistent growth limitation due to drought. For temperature, we found significant negative, but mostly unstable, correlations for the months from March to August from the mid-1970s onwards (Figure 8), which indicates reduced oak growth under high temperatures, notably from April to June.

The spatial correlation maps showed strong positive correlations between the first PC (i.e., PC1 scores) and April–September precipitation and drought (SPEI 08) over vast regions of the Middle East, including Kuwait, the south of Iraq, the east of Saudi Arabia, and the west of Iran (Figure 9). Higher correlations were found to drought, and, in general, the two field correlation maps indicate that the oak trees in the Zagros Mountains are subject to common regional hydroclimatic drivers.

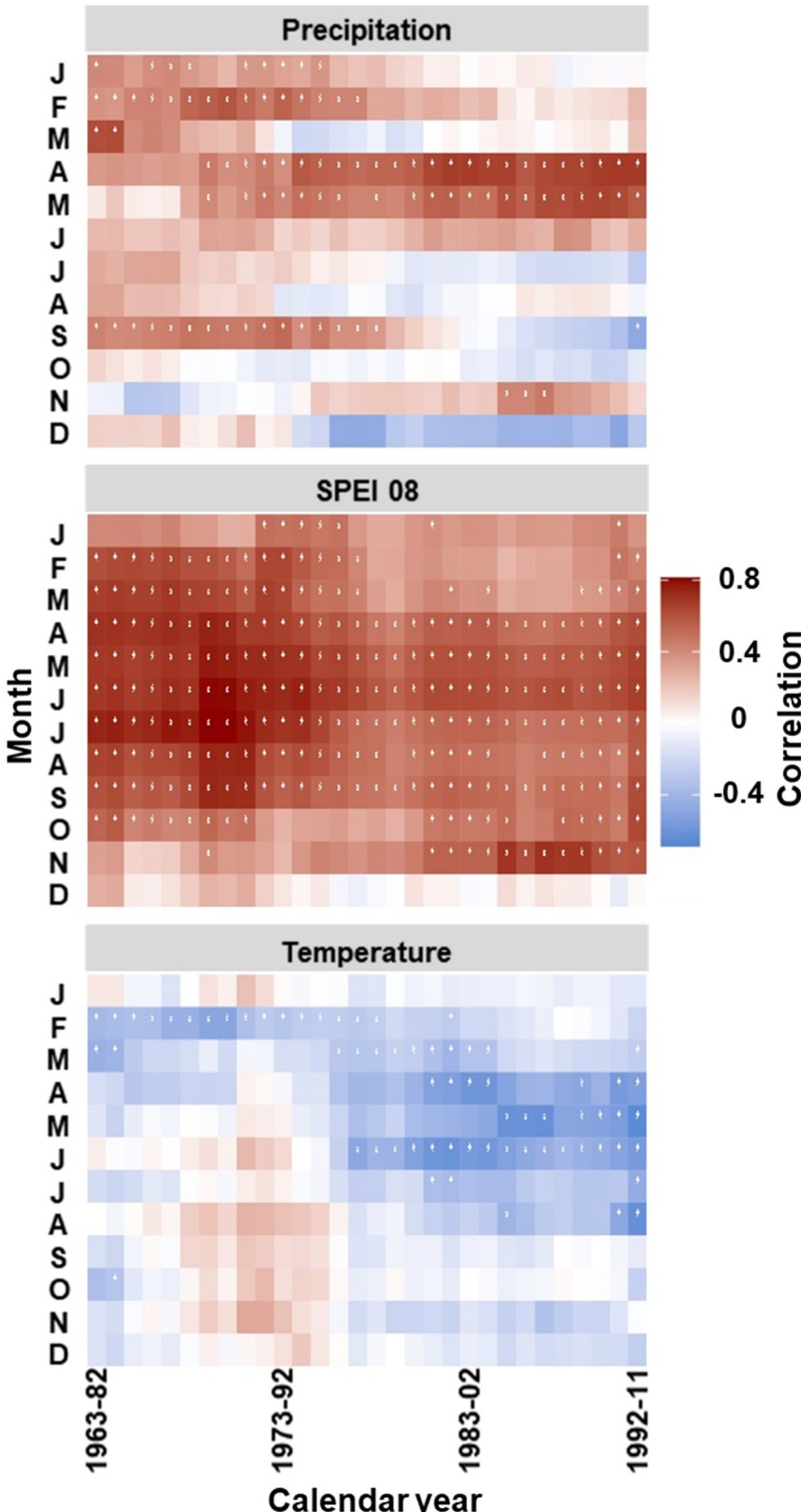

**Figure 8.** Moving correlations for 20-year windows between the first principal component (i.e., PC1) and the climate variables of total monthly precipitation, mean monthly SPEI 08, and temperature for the common 1963–2011 period. Moving correlation coefficients were calculated for each month of the year (from January to December). The color code represents correlation coefficients, and the white asterisks indicate significant correlations ($p < 0.05$).

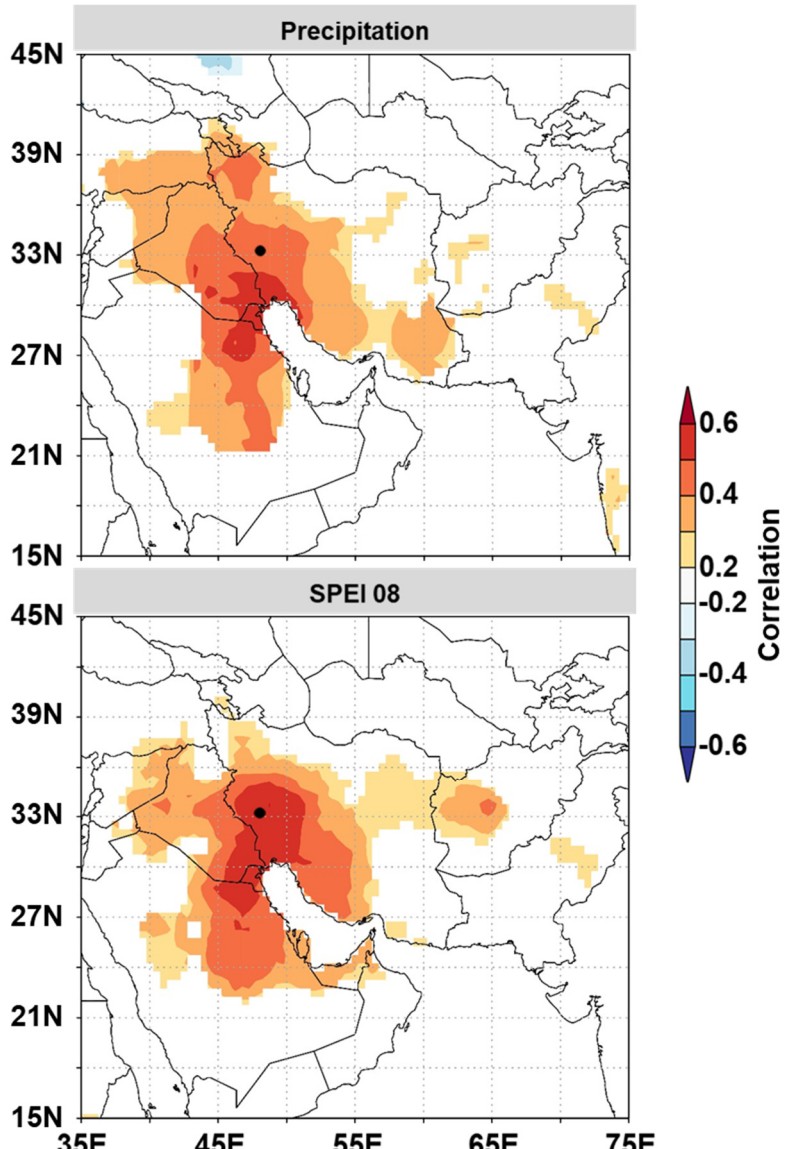

**Figure 9.** Spatial correlation maps between the first PC (i.e., PC1 scores) and April–September climatic variables using a resolution of 0.5° × 0.5° gridded cells for the common 1963–2011 period: upper map represents the correlation with precipitation (CRU TS4.04 land), and the lower map represents the correlation with drought (CSIC SPEI 08). The black dot indicates the location of the three sampling sites within the Central Zagros Mountain.

## 4. Discussion

### 4.1. Data Quality

*Q. brantii* is the dominant species in Lorestan's semi-arid oak forests in the Central Zagros Mountains. Due to legal constraints, our study was restricted to the use of trees felled by Lorestan's Department of Natural Resources for the purpose of limiting the spread of the oak decline in the region; thus, we could not deliberately apply our own criteria for the selection of study sites and sample trees. The analyzed trees were relatively young, which resulted in chronologies with a length of less than 70 years. Further, some of the samples from the agroforestry site were partly affected by forest decline symptoms. Oak decline has frequently been reported by researchers in Central Zagros [58], causing noticeable dieback within oak forests [29]. Even though the emergence of forest decline in the Zagros Mountains is stated to date back to 2000 [59], research at Mele-Shabanan, where our agroforestry site is located, declared oak trees to have been unaffected by this

phenomenon before 2010 [17]. Conversely, the common period of the three developed chronologies spans the years 1963–2011. Therefore, we expect no uncertainty of results due to some partially affected trees from the agroforestry site that are included in this study, which is proved by the almost identical growth pattern of the three chronologies (Figure 4b). However, future endeavours should focus on both, sampling healthy old trees as well as historical timber, to develop a long *Q. brantii* chronology for the Central Zagros Mountains.

Despite the notably different local environmental conditions and the large heterogeneity of the sample trees with respect to tree age and average growth rate, tree growth for the three sites is highly coherent, and highly significant, positive, inter-series correlations were obtained for all three TRW chronologies (Figure 4). The coherent growth pattern among the three chronologies confirms the strong common climate signals for all three sites, which is supported by the results of the PCA. A minor, hardly significant, share of only 17% of common overall variance likely remains for the differentiation between the three sites, providing evidence that the forest and agroforestry chronologies are slightly more similar than the silvopastoral chronology. PCA has been frequently used in dendroecological studies [60–65], as the significant PCs amplify climate signals [66] and give researchers a better insight into sites' heterogeneity [67].

### 4.2. Drought Signal

The strong positive loadings on PC1 reflect a robust common growth response of oak trees across the contrasting sites, which indicates that the radial stem growth of *Q. brantii* in the Central Zagros Mountains is synchronized through the same climate and climate factors, mainly by regional drought conditions. Furthermore, the high 83% share of the total chronology variance explained by PC1 provides evidence that effects of the site-specific, local environmental conditions, as well as effects of the stand density, tree age, tree size, and average tree growth rate on radial stem growth responses, are rather small. In fact, even though our sampling sites differ in management-type, topography, and altitude, and our sample trees differ considerably in age and growth history, only 17% of the total chronology variance was explained by PC2 and PC3.

The strong hydroclimate–growth correlations obtained for winter and spring indicate that the radial growth of *Q. brantii* is significantly dependent on and constrained by the water availability during that period. The positive effect of high winter precipitation on *Q. brantii* growth in Central Zagros has been reported by Parvaneh and Valipour [15]. In general, winter precipitation increases the available water sources and soil moisture storage. In contrast, late-spring or summer precipitation rarely ends up contributing to notable soil moisture storage or groundwater sources because much of the summer rain directly evaporates or runs off without infiltrating the soil [68,69]. Besides, in semi-arid climates, such as the Central Zagros Mountains, tree growth is extremely dependent on the rainfall, specifically in the early growing season (i.e., April) [14,21,30], where drivers, such as rising temperature, the onset of cambial activity, and leaves sprouting, provoke the trees towards more water consumption [70].

The moving window functions also confirmed the pivotal effect of winter (particularly January and February) and spring (here, April and May) precipitation on the growth. The consistent dependency of the tree growth on April and May precipitation from the late-1960s onwards coincides with the occurrence of decreasing precipitation rates in recent decades, as an indication of changing climate, for which the oak trees have encountered severe drought stress at the onset of cambial activity.

The moving window correlation analysis showed a trend of significant negative correlations for spring and early summer (June) temperature from the mid-1970s onwards, coinciding with a period of decreasing precipitation (Figures 2 and 8). The high air temperature in semi-arid Central Zagros means less soil moisture, which is a crucial limiting factor for cambial activity, and consequently for radial growth enhancement [14]. The negative temperature–growth relationship has been previously reported for Central [20,28], Southern [22] and Northern Zagros [71]. Finally, the results of the drought–growth moving

window correlation analysis, which integrates the effects of rising temperatures, decreasing precipitations, and soil moisture loss, point to a persistent growth limitation throughout the growing season (i.e., from April to September).

The period between water stress and placing the negative effects on tree growth varies markedly among forest types and tree families [72]. Our findings reveal stronger growth–drought associations for longer drought time scales of 8 and 24 months but a decline in growth–drought relationships at 36- and 48-month scales. Vicente-Serrano et al. [47] stated that SPEI at short time scales is predominately related to soil water content, while, at longer time scales, it is associated with variations in groundwater storage. That may explain the high drought–growth correlations during winter and spring (specifically April) at shorter SPEI time scales, where trees likely benefited from high precipitation rates. Furthermore, along with an increasing time scale, we found a progressive shift of growth sensitivity to drought from the winter to the summer–fall time window, while the drought signals recorded for spring exhibit a consistent trend.

Loadings on PC2 are bi-polar, exhibiting positive values for the silvopastoral chronology and negative values for agroforestry and forest. This contrasting growth response might be explained by the remarkably different topography, and, of course, forest management systems that have an extreme influence on both the forest stand density and trees' intraspecies competition; yet, only 10% of the total variance is explained by PC2.

Soil erosion is a common phenomenon in the Zagros region [73]; it is more prevalent in areas with silvopastoral management systems, which are subject to intensive grazing. At the steep rocky silvopastoral site, predisposition to soil erosion is extremely high, and the reduced vegetation cover is an additional sign of severe land degradation. Additionally, oaks' roots under these systems often are exposed to sunlight and thus are more sensitive to water stress than those at the other sites.

In direct contrast, agroforestry management systems are believed to noticeably enhance the vegetation cover, which subsequently counteracts soil erosion and provides a deeper root system [74]. The combination of deep- and shallow-rooting species (i.e., trees and crops) creates a spatial sharing of underground resources and contributes to larger resource utilization [75]. These systems maximize the exploitation of water in habitats with limited water supplies [76]. According to Anderson et al. [77], in agroforestry systems, as trees mature and develop their rooting system, the soil infiltration rate increases. Subsequently, groundwater sources would be more quickly replenished, and the trees, as well as, indirectly, the crops, benefit from more available water in deeper soil layers. For the forest site, the site conditions are more similar to the agroforestry site than silvopastoral. By the same token, a high infiltration rate and plasticity to water stress, and accordingly negative loadings on PC2, is expected for the forest site.

Our sites are located at an elevational range of 1326 to 1704 m (a.s.l.) and have an altitudinal difference from almost 150 up to 400 m. It is stated that at low-elevation sites radial stem growth is controlled by precipitation and related variables (e.g., soil moisture), whereas, at high elevations, it is controlled by temperature [78,79]. Our findings do not provide evidence to support this hypothesis.

### 4.3. Growth–Stand Density Relationships

The carbon allocation process brings trees both the competition capability for essential resources and the physiological adaptation in direct response to any environmental changes [80]. Root distribution is defined as trees' optimization strategy toward maximizing water access from soil layers [81]. To cope with the negative effects of competition, trees tend to develop their absorbing roots into deeper soil layers to enhance water accessibility and minimize root competition [82]. The highest AGR for the oaks at the silvopastoral site and the lowest for the forest are associated with the forest stand density and its impacts on trees' underground structures. We hypothesize that for the silvopastoral site with the lowest stand density, oaks benefited from unoccupied spaces in upper soil layers; thus, the stored water can easily be absorbed without the need to allocate more carbon to the root

system. Consequently, a high rate of carbon is allocated to aboveground structures, leading to the highest AGR for this site Alternatively, there is intense intraspecific competition and the lowest AGR at the forest site. Here, a higher rate of carbon should be allocated for vertical root distribution in order to access water and compensate for the negative effects of intraspecific competition. The developed roots enhance the root system maintenance costs [83] at the expense of aboveground tree development.

## 5. Conclusions

Our findings proved that the *Q. brantii* tree-rings conform with the expectation of a sound environmental proxy and can easily retain regional drought signals. Furthermore, as was spotlighted by the principal component analysis results, tree growth in the Central Zagros Mountains is controlled by regional climatic drivers (principally, precipitation and drought), and a mere 17% of the total variance was explained by differentiation among the sampling sites. The minimal modifying effect of the local environmental conditions and sample tree heterogeneity on the climate–growth relationship was also confirmed by the highly similar radial growth pattern of oak trees under different management treatments. Our results also revealed a consistent growth limitation over the growing season due to drought, which resulted from the decreasing precipitation in recent decades as an indication of climate change. However, further studies are needed that also include additional environmental parameters, such as soil moisture regime or soil fertility.

**Author Contributions:** Conceptualization, H.M., R.Y. and H.-P.K.; methodology, E.S., H.M., A.S., R.Y. and H.-P.K.; software, E.S., H.M., A.S. and W.T.; validation, E.S. and M.M.; formal analysis, E.S. and H.-P.K.; investigation, E.S., H.M. and J.S.; resources, R.Y., J.S. and H.-P.K.; data curation, E.S.; writing—original draft preparation, E.S., H.M., A.S., R.Y., W.T. and H.-P.K.; writing—review and editing, E.S., H.M. and H.-P.K.; visualization, E.S., A.S. and H.-P.K.; supervision, H.M., A.S., R.Y. and H.-P.K.; project administration, H.M., R.Y. and H.-P.K.; funding acquisition, R.Y. and H.-P.K. All authors have read and agreed to the published version of the manuscript.

**Funding:** This research received no external funding.

**Data Availability Statement:** Not applicable.

**Acknowledgments:** H.-P.K., H.M., M.M., E.S. and R.Y. acknowledge support by the European ERAS-MUS+ project between the University of Freiburg, Germany, and Isfahan University of Technology, Iran (agreement number 2020-1-DE01-KA107-005644). The authors thank Felix Baab and Clemens Koch for technical support during laboratory work, and acknowledge support by the Open Access Publication Fund of the University of Freiburg.

**Conflicts of Interest:** The authors declare no conflict of interest.

## Appendix A

**Table A1.** Mann–Kendall statistics for precipitation, temperature, and drought (represented as SPEI 01 to SPEI 48) trends for the individual months (from January to December), short (i.e., A–S) and long (i.e., M–O) vegetation period, and for the whole year over the common period 1963–2011.

|  | Temp. | Precip. | SPEI 01 | SPEI 03 | SPEI 04 | SPEI 06 | SPEI 08 | SPEI 12 | SPEI 16 | SPEI 24 | SPEI 36 | SPEI 48 |
|---|---|---|---|---|---|---|---|---|---|---|---|---|
| JAN | **−2.25 \*** | −0.59 | −0.44 | −0.70 | −0.44 | −0.54 | −0.49 | −1.23 | −0.42 | −1.39 | **−2.4 \*** | **−2.22 \*** |
| FEB | −1.1 | −1.11 | −1.08 | −0.46 | −0.78 | −0.75 | −0.77 | −1.41 | −1.01 | −1.61 | **−2.44 \*** | **−2.51 \*** |
| MAR | **−2.39 \*** | 0.13 | 0.00 | −0.85 | −0.37 | −0.58 | −0.68 | −1.65 | −1.63 | −1.58 | **−2.4 \*** | **−2.46 \*** |
| APR | **−2.21 \*** | −0.11 | −0.42 | −0.85 | −1.03 | −0.53 | −0.42 | −0.92 | −1.51 | −1.58 | **−2.46 \*** | **−2.27 \*** |
| MAY | −1.46 | **−2.09 \*** | −1.47 | −0.96 | −1.39 | −1.41 | −0.96 | −0.89 | −1.78 | −1.68 | **−2.34 \*** | **−2.42 \*** |
| JUN | −0.86 | 0.80 | −1.18 | −1.18 | −1.16 | −1.75 | −0.89 | −0.94 | −1.59 | −1.80 | **−2.35 \*** | **−2.39 \*** |
| JUL | −1.72 | 1.14 | −0.44 | −1.59 | −1.13 | −1.41 | −1.41 | −1.03 | −1.41 | −1.73 | **−2.37 \*** | **−2.39 \*** |
| AUG | −1.21 | 1.13 | −0.23 | −0.72 | −1.54 | −1.06 | −1.63 | −1.04 | −1.15 | −1.72 | **−2.22 \*** | **−2.32 \*** |
| SEP | **−2.1 \*** | −0.07 | 0.04 | −0.13 | −0.77 | −1.11 | −1.54 | −0.96 | −0.99 | −1.80 | **−2.23 \*** | **−2.27 \*** |
| OCT | **−2.4 \*** | 0.13 | −0.03 | −0.42 | −0.53 | −1.73 | −1.58 | −1.41 | −1.32 | −1.68 | **−2.27 \*** | **−2.59 \*\*** |

**Table A1.** *Cont.*

|  | Temp. | Precip. | SPEI 01 | SPEI 03 | SPEI 04 | SPEI 06 | SPEI 08 | SPEI 12 | SPEI 16 | SPEI 24 | SPEI 36 | SPEI 48 |
|---|---|---|---|---|---|---|---|---|---|---|---|---|
| NOV | −4.45 *** | 0.54 | 0.61 | −0.04 | 0.08 | −0.18 | −0.59 | −1.75 | −1.32 | **−2.08 *** | **−2.37 *** | **−2.58 **** |
| DEC | **−2.1 *** | −0.49 | −0.53 | 0.06 | 0.06 | −0.20 | −1.13 | −1.72 | −1.37 | **−2.15 *** | **−2.8 **** | **−2.73 **** |
| A-S | −1.6 | −0.85 | −0.97 | −1.22 | −1.28 | −1.25 | −1.15 | −0.92 | −1.53 | −1.78 | **−2.28 *** | **−2.39 *** |
| M-O | −1.82 | −1.04 | −1.54 | −1.47 | −1.54 | −1.66 | −1.20 | −0.85 | −1.56 | −1.85 | **−2.27 *** | **−2.46 *** |
| Annual | **−2.32 *** | **−2.01 *** | −1.49 | −1.66 | −1.73 | −1.54 | −1.16 | −1.25 | −1.56 | −1.90 | **−2.47 *** | **−2.65 **** |

Significance levels: * $p < 0.05$, ** $p < 0.01$, and *** $p < 0.001$.

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
