# Peer review of "Regional Drought Conditions Control Quercus brantii Lindl. Growth within Contrasting Forest Stands in the Central Zagros Mountains, Iran"

_forests, doi:10.3390/f13040495_

Round 1

Reviewer 1 Report

The study provide evidence that hydroclimatic conditions control tree-ring formation in this region, which highlights the great potential for combining historical oak samples and living trees from different forest stands to generate multi-centennial tree-ring based hydroclimate reconstructions.

Line50: The author points out that some studies have analyzed the climate sensitivity of oak tree ring width, and can give some examples, please complete.

Line337-338:It is suggested to add the question that the study is expected to address at the end of the text as a conclusion.

Line422-424:From “vegetation” to “site”, the font size is inconsistent with the context, please modify.

Compared with previous research, what is the innovation of this experiment, please add. Please point out the limitations and advantages of this research method.

I am not very clear about the experimental assumptions of this article, and what is the theoretical basis?

Regarding this research direction, what problems are still unresolved, please add.

Author Response

Dear Reviewer,

Please see our rebuttals to your comments in the attachment. 

Remark to differences between the line number references that the reviewers have used and the line numbers that we have used in the rebuttals: The reviewers requested some lines and references of the manuscript be omitted or summarized. Also, a few new references according to their comments were cited. Furthermore, some parts of the introduction and discussion went through a bit of restructuring to fulfill a number of comments. All this had caused almost a thorough line relocation throughout the manuscript, which led to those differences.

Reviewer 2 Report

The manuscript is the case study of Persian oak radial growth and its climatic response in stands with varying local conditions and human activities. It is of mild interests for dendrochronologists, since (as presented in the Introduction) this species and study area is already well-studied.

Abstract well represents contents of tу manuscript and does not repeated in conclusion section.

Introduction reflects relevance, aim and tasks of the study, and state of art adequately. However, its structure could be improved. I recommend to move sentences about Zagros forests from L87-94 to the end of higher paragraph about biodiversity hotspots in L49, then put oak species description from L80-87 in the beginning of the next paragraph before listing of its researches.

Materials and Methods are described mostly in sufficient details. But  presented in text and Table 1 tree density is too low in the first two sampling sites to consider them as forest stands, they are rather pasture / field with scattered trees. With such low density, its unevenness (homogeneous scattering of single trees, or their locations as groups / rows between herbaceous patches) can impact degree of competition severely, so tree location pattern should be at least briefly described. Also, please state approximate timeframe of insect/fungi impact on sampled trees before their felling. And negative temperature trend in Figure 2 and Table S1 contradicts discussed later warming (L389-390).

Results are described in consecutive and logical manner, but there are some recommendations on improvement of illustrative material. Most of content of Table 1 is repeated in text; you should remove these numbers from text or preferably add sites' coordinates in text and omit this table totally. In the Figure 1, map of just Lorestan area with separate markers for each site and for meteostation should be more informative (general country map can be included as an insert. In regards to Table 2, numbers from it also should be deleted from text (in L24-228). 

Discussion contains mostly valid arguments. But paragraph in L353-361 belongs rather to Materials and Methods. Looking on SPEI08, you forget that it already contains eight months, so its integration from April to September (e.g., L402) actually refers to period from previous September to current one. Absence of full gradient from moisture- to temperature-limited tree growth mentioned in L437-441 is to be expected within just 400 m of elevation difference in the warm region. All your sites are within small part of this ecological gradient where tree growth is more or less limited by moisture deficit. Then, with 92 trees/ha even in the densest site, competition between trees should not be "intense" (L455), although I agree that it is more evident in the forest site. And not only roots, but height growth should be more necessary for denser stand, because competition for light instigates "race for height" between neighboring trees.

In regards to tree age issue due to restricted sample choice and protected nature of the Persian oak forests, did you consider asking for permission of cautious sampling from living trees with increment borer? If sampling is performed during tree dormancy period to avoid excessive sap leakage, borer is disinfected after each tree, and holes from borer are sealed with gummy substance immediately, the possibility of danger to trees diminishes greatly. We, for example, were able to persuade our local natural reserves to permit tree sampling on their territories on such conditions (with presence of their employees of course).

References and their citations should be re-formatted according to journal requirements (citations as numbers in square brackets like [1], [3–6], or [2,7–10]; reference list should be in order of its citation, formatting of references as shown here: https://www.mdpi.com/authors/references and here: https://www.mdpi.com/journal/forests/instructions#references).

Minor comments:

L2 and many other times throughout the text. Write Latin species names in italics.

L5. delete "and", and add comma after 4th author.

L7-17. Check missing emails, and show which is whose.

L18. Add email of Mahsa Mirzakhani if they are too corresponding author (journal allows two corresponding authors)

L44,72,74 and several more cases. In word combinations like "tree–environment interaction" and other similar descriptions of relationships between so and so, please use dash (–) instead of hyphen (-) to demonstrate that it is not one word.

L82. Correct formatting of the number.

L85. "economically" instead of "economical".

L107. Cit just Fig.1, not panels a-d.

L108. Delete comma.

L127, 162,164,Table 2, and Fig.4b. Check common period, years don't match.

L209. State that CRU TS are temperature and precipitation data, not just climate.

L216. Cite Table 2 here, and delete table citations from L219, 220.

L220. Lowest average AGR?

L234-237. I recommend to write simpler: "PC2 differentiates between negative loading for forest and agroforestry and positive loading for silvopastoral. PC3 differentiates between negative loading for agroforestry and silvopastoral and positive loading for forest."

Fig. 5. Missed panel labels (a, b) within figure.

L261. "Figure" in single form.

L266. Delete repetition of figure citation.

L271-272. There should be plural form: "the responses ... are strongest".

Figure 7 lacks common title before description of panels. Perhaps, "Comparison of relationships between the principal components (i.e. PC1, PC2 and PC3) and climatic variables: a) mean April-September SPEI of one to 48-month time scales, and b) April-September means of..." Also better use p<0.05 and p<0.01 in the caption too (consistent with legend).

L292. Perhaps, "extent"?

L367. Use single form "provides".

Overall, the manuscript should be corrected attentively before it can be published.

Author Response

(The authors gave the same response as above.)

Reviewer 3 Report

General Comments:

This article presents a dendroecological study in the Mediterranean part of Iran. As it is a research paper, we may expect this to be a novel approach. If the paper is extending similar earlier work, then references should be made. Some papers about the effect of climatic parameters on oak decline and Persian oak trees should be added to the text, for example:

  • Attarod, P., Rostami, F., Dolatshahi, A., Sadeghi, S. M. M., Amiri, G. Z., & Bayramzadeh, V. (2016). Do changes in meteorological parameters and evapotranspiration affect declining oak forests of Iran?. Journal of Forest Science, 62(12), 553-561.

Also, some great papers ignored by authors:

Arsalani, M., Pourtahmasi, K., Azizi, G., Bräuning, A., & Mohammadi, H. (2018). Tree-ring based December–February precipitation reconstruction in the southern Zagros Mountains, Iran. Dendrochronologia49, 45-56.

Arsalani, M., Griessinger, J., Pourtahmasi, K., & Bräuning, A. (2021). Multi-centennial reconstruction of drought events in South-Western Iran using tree rings of Mediterranean cypress (Cupressus sempervirens L.). Palaeogeography, Palaeoclimatology, Palaeoecology, 567, 110296.

Specific Comments:

- Line 21 and rest parts: Scientific name should be italic.

- Line 23: Change "1.326 to 1.704 m a.s.l." to "1,326 to 1.704 m a.s.l". Change it for the following parts of manuscript. 

- Line 24: "low for forest site". It is not clear to me. 

- Line 25: "Common climatic signal", like what? 

- Lines 39-40: "Today.... (Lindner et al., 2010)". It's 2022, and you need to use an updated citation. 

- Line 46: To my knowledge, the central part of Zagros is not a biodiversity hotspot. 

- Line 46 and Line 47: There is no coherency and connectyion between these lines. 

- Line 63: Persian oak or Quercus brantii. Use one of them throughout the manuscript. 

- Line 65 and Line 66: There is no coherency and connectyion between these lines. Remove this sentence: "Management intensity can place direct or indirect effects on forest ecosystems (Yousefpour et al., 2012) resulting in changes in the vegetation composition (Moradi et al., 2012), which in turn can influence the trees’ response to weather and climate".

- Line 77: Pinus pinaster should be italic.

- Line 79: Add some references from the Zagros forest of Iran. 

- Line 89: Again, it is not true about the central or southern part of Zagros. 

- Lines 95-97: Very poor research necessity. Need to reconsider why your research is important. 

- Last paragraph of introduction: Add research hypothesis and research question.

- Table 1: How about the form of trees? Is it coppice or high? Long and Lat should be stated in more high resolution. 

- Figure 1: I would like to see a detailed map about the distribution of three sites in Khoramabad city. 

- Lines 134-136: "The discs were collected from trees which were felled by the Lorestan’s Department of Natural Resources between years 2011–2015 to prevent a spread of oak dieback...". Are you sampled from unhealthy trees? How oak decline can affect your results? I did not see any interpretation for this in the discussion, and it is cause for some uncertainty of your results. 

- Line 140: Need a citation.  

- Line 162: According to past studies, since 2002, Zagros forest has a problem with oak decline. So, I am really recommend to limit your analysis up to 2002 to exclusive oak decline phenomena. 

Author Response

(The authors gave the same response as above.)

Round 2

Reviewer 3 Report

Good job! The revised version has been greatly improved.